## Methods and techniques

**Subject Area:**
developmental biology/cellular biology/
biotechnology/genetics/genomics/molecular
biology

gastrulation, avian embryos, transcriptomics,
cell polarity, blastoderm, blastodisc

**Author for correspondence:**
Claudio D. Stern
e-mail: c.stern@ucl.ac.uk

†Present address: Institute of Cancer Research,
123 Old Brompton Road, London SW7 3RP, UK.

Electronic supplementary material is available
online at https://doi.org/10.6084/m9.figshare.
c.4829343.

# Molecular anatomy of the pre-primitive-streak chick embryo

Hyung Chul Lee, Hui-Chun Lu, Mark Turmaine, Nidia M. M. Oliveira, Youwen Yang†, Irene De Almeida and Claudio D. Stern

Department of Cell and Developmental Biology, University College London, Gower Street, London WC1E 6BT, UK

  HCL, 0000-0001-5438-1900; CDS, 0000-0002-9907-889X

The early stages of development of the chick embryo, leading to primitive streak formation (the start of gastrulation), have received renewed attention recently, especially for studies of the mechanisms of large-scale cell movements and those that position the primitive streak in the radial blastodisc. Over the long history of chick embryology, the terminology used to define different regions has been changing, making it difficult to relate studies to each other. To resolve this objectively requires precise definitions of the regions based on anatomical and functional criteria, along with a systematic molecular map that can be compared directly to the functional anatomy. Here, we undertake these tasks. We describe the characteristic cell morphologies (using scanning electron microscopy and immunocytochemistry for cell polarity markers) in different regions and at successive stages. RNAseq was performed for 12 regions of the blastodisc, from which a set of putative regional markers was selected. These were studied in detail by *in situ* hybridization. Together this provides a comprehensive resource allowing the community to define the regions unambiguously and objectively. In addition to helping with future experimental design and interpretation, this resource will also be useful for evolutionary comparisons between different vertebrate species.

## 1. Introduction

Like most amniotes, the avian egg is fertilized internally within the mother's reproductive system. In the chick, the early zygote spends about 20 h inside the mother's reproductive system while initial meroblastic cell divisions take place, before being laid as a flat blastodisc, estimated to contain about 20 000 cells [1,2]. This blastodisc comprises a central area pellucida (AP) and a peripheral, extraembryonic area opaca (AO), separated by a thin ring, the marginal zone (MZ). After laying, about 15 h elapse before the beginning of gastrulation, marked by the appearance of the primitive streak, the site of mesoderm and endoderm formation, at one edge of the AP. The primitive streak is the first obvious morphological sign of bilateral symmetry. Prior to primitive streak formation, the blastodisc undergoes very extensive global cell movements (called 'Polonaise' in the epiblast, and other movements in the lower layer) [3–11], which elongate the domain of primitive streak formation along the future midline. The polarity of the blastodisc remains plastic until the appearance of the streak, which is most dramatically demonstrated by cutting the embryo into fragments: as long as each fragment contains a portion of the inner AP and of the MZ, they can all give rise to a primitive streak and thereafter to a complete embryo [12–14].

Despite its apparent morphological simplicity, the blastodisc comprises a number of regions and cell populations. These have been studied for more than two centuries since Pander and von Baer first discovered the 'germ layers' [15,16], but the terminology to describe them has often changed over time, between countries and even between different schools of embryologists.

royalsocietypublishing.org/journal/rsob Open Biol. **10**: 190299

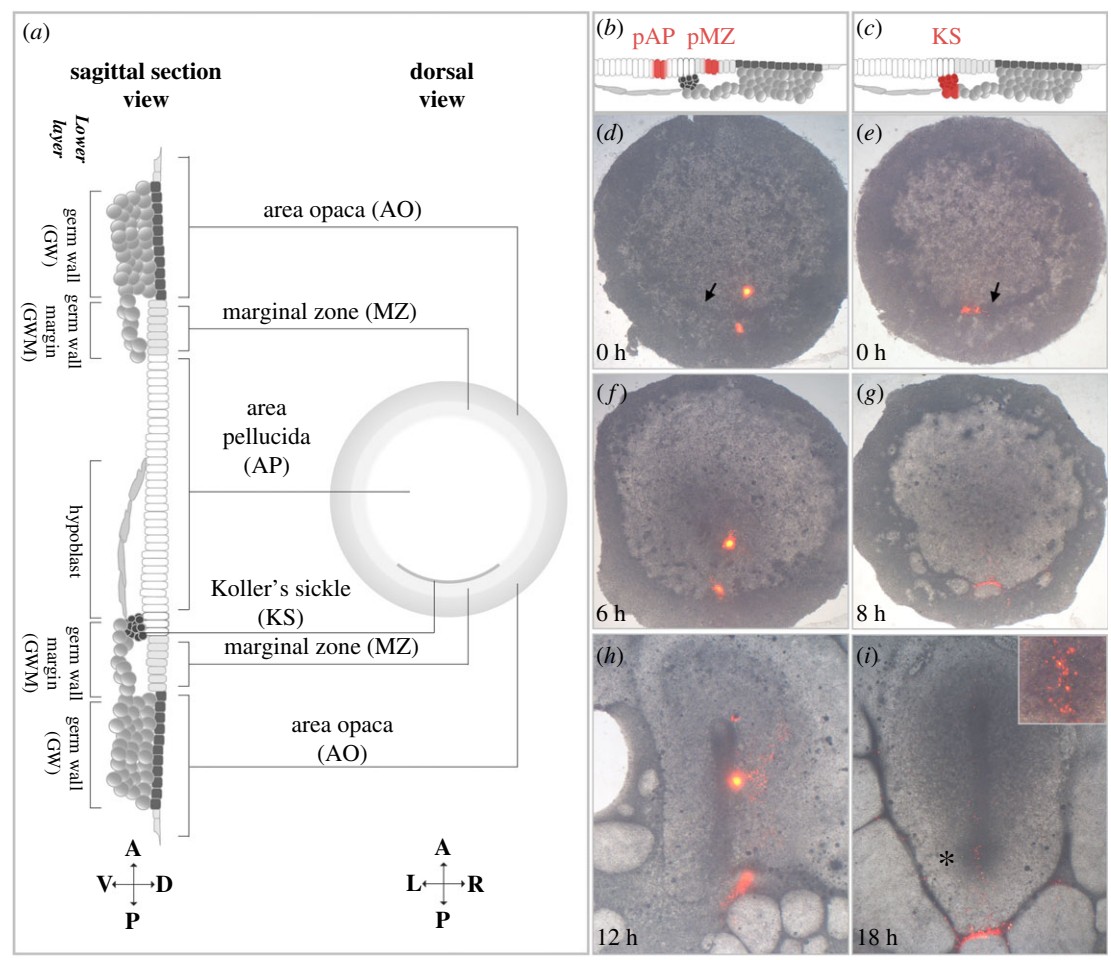

**Figure 1.** Anatomy of pre-primitive streak chick embryos. (*a*) Anatomy of a pre-primitive-streak stage chick embryo in dorsal view (right) and as mid-sagittal section (left). The cardinal points below each diagram show the orientation: A, anterior; P, posterior; L, left; R, right; D, dorsal; V, ventral. (*b–i*) Fate-mapping cells of the posterior area pellucida (pAP) and posterior marginal zone (pMZ) (*b, d, f, h*), and of KS (*c, e, g, i*) using DiI labelling at different time points as indicated. Cells labelled in KS or pAP contribute to the primitive streak, while those labelled in the pMZ remain extraembryonic. Arrows in (*d,e*): KS. Inset in (*i*): magnified view of posterior primitive streak region (asterisk).

Now that there is renewed interest in many of the processes that occur at these stages, and given that there are excellent tools for objective and detailed description of morphological, molecular and functional characteristics of cells and cell populations, an opportunity arises to unify the terminology in a fully portable way as well as to provide a clear translation of the main terms in the historical literature to make this more accessible to modern readers. Here, we undertake this task. We combine confirmatory cell labelling with morphological studies by scanning electron microscopy (SEM), a survey of changes in cell polarity using immunocytochemistry, an analysis of the transcriptome of 12 defined regions of the blastodisc and validation of the key markers by *in situ* hybridization. These data are combined to define the characteristics that allow the regions to be identified by several criteria, to accommodate different types of experiments. The transcriptome and newly identified markers should also be useful for comparing similarities and differences in cell populations and molecular properties among different species of vertebrates.

## 2. Terminology and staging

To guide the reader, we first present a brief description of the anatomy (figure 1*a*) along with a summary of the changing terms used to refer to different tissues and cell populations over time (table 1).

The blastodisc can be divided into three concentric regions: an outer AO, an inner AP and an intermediate, narrow ring situated between the former two called the MZ. The boundary between AP and MZ, at the future posterior part of the embryo, is marked by a crescent-shaped ridge called Koller's sickle (KS) [25] (figure 1*a*). Dorsally, a largely single-cell-thick layer (epiblast) is continuous across all three regions. Ventral to this, there are several cell layers that differ according to region and stage of development. These are shown in figure 1*a* (see also table 1). They include the germ wall (GW) (several layers of large and yolky cells continuous with the yolk under the epiblast of the AO and attached to its ventral [deep] aspect), the hypoblast (a single cell sheet of cells with intermediate amounts of yolk under the epiblast of the AP, which forms a cell sheet gradually from posterior to anterior) and the germ wall margin (GWM) (similar to the GW, underlying the epiblast of the KS and the MZ but not attached to the epiblast).

The approximately 15 h of development between laying and primitive streak formation are most clearly marked by gradual changes of the hypoblast, which starts as a loose set of 'islands' scattered against the ventral surface of the AP epiblast (thought to arise before laying by polyingression from the epiblast [26]), and these then gradually fuse with each other in a posterior-to-anterior direction. The degree to which the hypoblast has formed a sheet is the major criterion

**Table 1.** Modern and other older anatomical terms of the early chick embryo.

| modern terminology | description | comments | other older names | selected references |
|---|---|---|---|---|
| epiblast | uppermost cell layer | | ectophyll, ectoblast, ectoderm | Lillie [2] and Romanoff [1] |
| hypoblast | initial extraembryonic lower layer under area pellucida (initially separate islands fuse to form a continuous sheet of cells) | extraembryonic fate: germinal crescent, yolk sac stalk | endophyll ('primary hypoblast'; Vakaet, Stern 1990), sickle endoblast ('secondary hypoblast'; Callebaut) | Vakaet [11], Eyal-Giladi & Kochav [17], Bellairs et al. [18], Stern [19] and Callebaut & van Nueten [20] |
| endoblast | posterior GWM-derived extraembryonic lower layer; displaces hypoblast sheet | extraembryonic, yolk sac endoderm | entoblast (Vakaet), junctional endoblast (Vakaet, Stern) | Vakaet [11], Callebaut & van Nueten [20], Stern [19] and Bachvarova et al. [21] |
| germ wall | lower layer of area opaca | peripheral extraembryonic endoderm (large yolky cells) | area opaca endoderm or area opaca endoblast (Vakaet, Stern) | Stern [19] and Bachvarova et al. [21] |
| germ wall margin | germ wall portion underlying the marginal zone (often protrudes to cover Koller's sickle) | peripheral extraembryonic endoderm (large yolky cells)—posterior portion generates endoblast | deep portion of marginal zone (Stern) | Stern [19] and Bachvarova et al. [21] |
| marginal zone (epiblast only) | intermediate ring of epiblast between area opaca and area pellucida; inner boundary is Koller's sickle | extraembryonic epiblast—no contribution to embryonic tissues | epiblast of the marginal zone (Eyal-Giladi), a superficial portion of the marginal zone (Stern) | Spratt & Haas [10], Eyal-Giladi & Kochav [17] and Bachvarova et al. [21] |
| definitive endoderm | embryonic endoderm; derived from tip of the primitive streak | fate: gut lining, other endodermal organs | entoderm, definitive endoblast, gut endoderm, endoderm | Bellairs [22], Stern & Ireland [23] and Kimura et al. [24] |
| Koller's sickle | crescent-shaped ridge located posteriorly; marks boundary between posterior area pellucida and marginal zone | contributes to primitive streak, some endoderm and mesoderm | Rauber's sickle (Callebaut) | Koller [25], Callebaut & van Nueten [20] and Bachvarova et al. [21] |
| mesoderm | | fate: cardiovascular, musculoskeletal, etc. systems | mesoblast | Lillie [2] and Romanoff [1] |

royalsocietypublishing.org/journal/rsob Open Biol. **10**: 190299

used for staging embryos at these stages, as defined by Eyal-Giladi & Kochav [17] who divided pre-primitive streak stages into 14 stages (EGK I–XIV), replacing stages 0 and 1 of Hamburger & Hamilton [27]. The egg is laid at EGK X, when the lower surface of the AP contains a variable number of clusters (islands) of hypoblast precursors. From EGK XI–XIII, these islands converge to form a sheet starting posteriorly, adjacent to KS. At EGK XI, it has covered about the posterior third of the AP, half of the AP at EGK XII and all the AP at EGK XIII–XIV (electronic supplementary material, figure S1). EGK XIV is characterized by the appearance of a thickened 'posterior bridge' [17] behind KS but this is not seen in all embryos or in all strains of fowl (unpublished observations), so we have not included this structure in our analysis. The Hamburger & Hamilton system (HH, in Arabic numerals) starts at HH2, characterized by the appearance of the primitive streak, initially as a short triangular shape at the posterior AP, protruding forwards from KS. In some embryos, this stage is absent, the primitive streak first appearing as a parallel-sided structure (stage HH3) [5,6,27].

## 2.1. Marginal zone

The intermediate marginal ring of epiblast (MZ) separating the peripheral AO from the inner AP is critically important for the formation of the embryonic axis, as first revealed by the pioneering studies of Spratt [10] and subsequently by work from Eyal-Giladi and others [17,21,28–32]. Although some studies (e.g. [19]) used the term 'MZ' to refer to the whole thickness of this intermediate region (epiblast plus GWM), the current convention is to define the MZ proper following the original criteria of Eyal-Giladi's group, and comprising only the epiblast component. Fate maps have shown that this region of epiblast does not contribute any cells to the embryo proper—the cells remain in a peripheral position and the primitive streak arises from the adjacent posterior AP epiblast [21]. Here, we follow the main criteria of Eyal-Giladi: the MZ is easiest to define posteriorly, where its inner boundary corresponds to the position of KS, and its outer boundary is the place where the deep yolky cells of the GW are attached to the epiblast of the AO. At the MZ, the GWM is easily detached from the epiblast posterior to the sickle. This belt of epiblast is projected all the way around the embryo to the anterior side to define a ring-shaped region with the same thickness (figure 1a). Importantly in all functional definitions, note that the MZ does not include the KS itself. Cells of the MZ do not contribute to the primitive streak, while some KS cells do (figure 1b–i and [21,33,34]).

## 2.2. Lower layers

The cellular composition of the lower layer changes over time and includes several different cell populations. The names used to refer to these populations have differed much more than for other cell layers, partly because the fact that the pre-primitive-streak lower layer does not contribute significantly to the definitive (gut) endoderm was not discovered until the 1950s [22,35–38], and partly because of different terminology employed by the Belgian, Israeli, UK and US schools of avian embryologists in the 1960s–1980s. Table 1 provides a glossary to compare the terms, although the precise definitions of these components also varied somewhat.

The sagittal section schematic in figure 1a illustrates the different components now generally recognized. Peripherally, the AO epiblast is underlain by the GW, the MZ by the GWM and the AP initially by hypoblast (see above). These three cell populations can be distinguished by morphology and fate. Both GW and GWM consist of relatively large yolky cells; both contribute to a chorion-like extraembryonic region [21]. In addition, a subpopulation of GWM cells at the posterior part of the embryo (expressing the HNK1 epitope) contributes to a cell layer that migrates centripetally towards the AP and starts to displace the hypoblast at stage EGK XIII–XIV [19,39–42]. This new AP deep layer is referred to as endoblast (junctional endoblast and other names in the older literature; table 1). Collectively, the hypoblast and endoblast are referred to as lower layer, following Vakaet's group's convention [11,21,23,39,40,43–45]. The lower layer of the pre-primitive streak embryo as a whole is comparable to the visceral endoderm of the mouse, where the hypoblast is functionally and molecularly analogous to the anterior visceral endoderm [41,46].

The hypoblast eventually becomes displaced further anteriorly by the ingression of epiblast derived cells to form the definitive (gut) endoderm at the tip of the primitive streak [10,11,24,35,38,41]. The hypoblast ends up in the anterior crescent-shaped arc known as the germinal crescent because it carries the primordial germ cells to this location [47–51].

# 3. Results

## 3.1. Cell morphology and polarization in the pre-primitive-streak stage embryo

We started by analysing the cell shape changes that take place as the epiblast transitions from a relatively loose arrangement of cells to a more distinct columnar epithelial organization—this is somewhat equivalent to a mesenchyme-to-epithelium transition [52]. First, we used SEM of sagittal sections (figure 2a–e). The aspect ratio (AR) between the major and the minor axis of each epiblast cell was investigated by manual segmentation and the ratio represented as a pseudo-colour heat map (figure 2a–e and f–k). In the AP, the AR increases gradually and significantly during development (figure 2a–d). On the other hand, cells of the AO maintain their cuboidal morphology, and the cells are smaller than in the AP at all stages (figure 2a–c,e). Although we did not observe significant differences in AR between anterior and posterior regions in sagittal sections (figure 2a–c), transverse sections through anterior and posterior regions of the AP were examined and compared (figure 2f–k). At EGK X, no significant difference in AR was observed between the two regions (figure 2f,g,j). However, by EGK XIII, cells of the posterior epiblast are more elongated (columnar) than their anterior counterparts, especially near the midline (figure 2h,i,k). This may be an early indication of the impending start of primitive streak formation from the posterior epiblast [5,6].

Next, we further examined the process of cell polarization by immunostaining for cell polarity markers PAR3, PKCζ and GM130 (figure 2l–t). Both PAR3 and PKCζ are localized apically in the cytoplasm and membrane even at EGK X–XI, and this becomes more marked with development for PKCζ (but not for PAR3) (figure 2l,m,o,p,r,s). Both markers are

royalsocietypublishing.org/journal/rsob    Open Biol. **10**: 190299

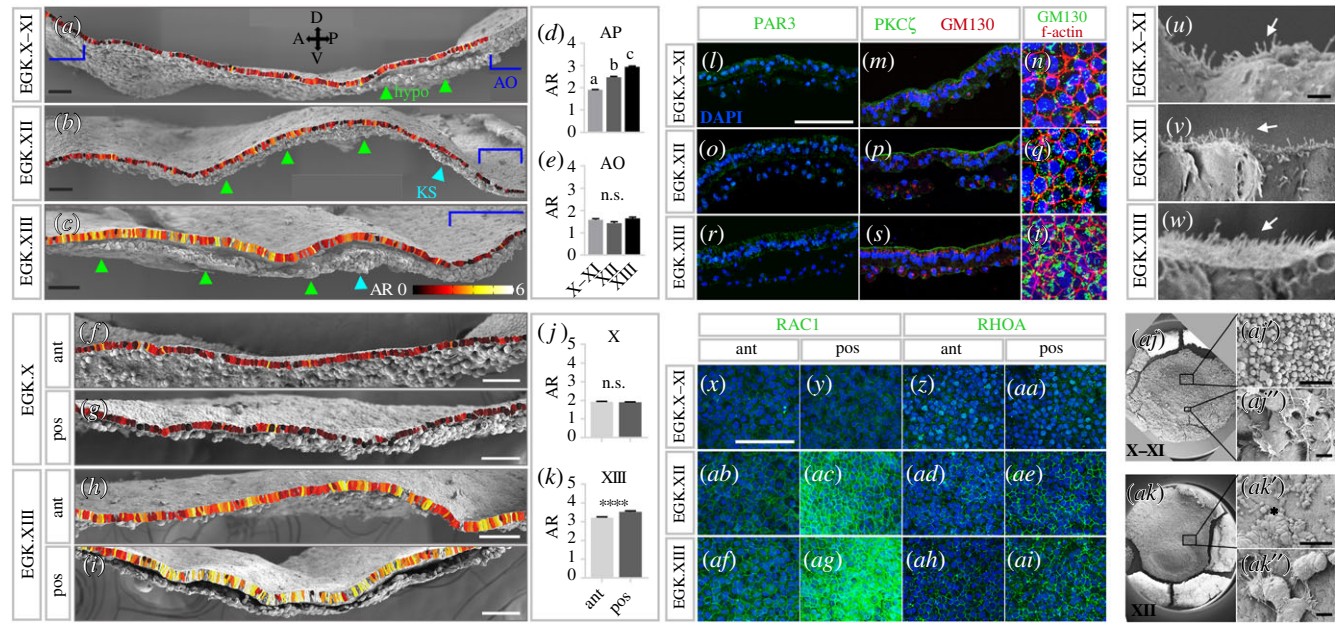

**Figure 2.** A wave of gradual cell polarization traverses the epiblast before primitive streak formation. (*a–c*) Analysis of the AR between the major and minor axis of epiblast cells measured from scanning electron micrographs represented as a heat map. AO (blue brackets), hypoblast (hypo, light green arrowheads) and KS (cyan arrowheads) are indicated. (*d–e*) Quantification of average AR in the AP and AO shows an increase of AR in the AP (*d*) with development, while no such change is seen in the AO (*e*). Mean ± s.e.m. shown; one-way ANOVA followed by Tukey's post-test: n.s., not significant; a–c indicate significant difference at $p < 0.0001$ (further details in electronic supplementary material, table S2). (*f–i*) AR of anterior (ant) and posterior (pos) regions compared at EGK X and EGK XIII. (*j–k*) AR between the two regions at EGK X and XIII. No significant difference at EGK X (*j*), but significantly higher AR in the posterior region at EGK XIII (K). **** $p < 0.0001$; unpaired *t*-test (further details in electronic supplementary material, table S2). (*l–t*) Polarity markers show progressive apical localization during development. PAR3 and PKCζ are first concentrated in nuclei. The Golgi marker GM130 becomes localized from EGK XIII (*s–t*). Phalloidin staining for f-actin marks cell boundaries. (*u–w*) The number of microvilli on the apical surface (arrows) increases gradually during development. (*x–ai*) Rho family small GTPases RAC1 and RHOA gradually increase in expression and become concentrated posteriorly with development. (*aj–ak*) Ventrally, SEM shows the gradual formation of the hypoblast sheet (*aj–aj″* and *ak–ak″*). At EGK X–XI, posterior cells are interconnected (*aj″*), whereas anterior cells are dispersed (*aj′*). At EGK XII, the hypoblast sheet covers half of the AP diameter (*ak*). Higher magnification (*ak′, ak′*) shows numerous filopodia and lamellipodia. Scale bars: 100 µm in (*a–c*), (*f–i*), (*l*), (*m*), (*o*), (*p*), (*r*), (*s*), (*x–ai*), (*aj′*), (*ak′*); 10 µm in (*n*), (*q*), (*t*), (*aj″*), (*ak″*); 2 µm in (*u–w*).

also detected in nuclei initially (figure 2*l,m*), but this decreases gradually. In polarized cells, the position of the Golgi body is often used as an indicator of apical-basal polarity by defining a Golgi-nucleus axis (the Golgi is positioned apically to the nucleus in epithelia) [53]. To check this, we immunostained sections for the Golgi marker GM130. GM130-immunoreactivity was initially random within epiblast cells, but from EGK XIII, Golgi ribbons are observed apically to the nucleus (figure 2*m,n,p,q,s,t*). Concomitant with this gradual cell polarization, the number and length of microvilli projecting from the apical surface increases gradually during development (figure 2*u–w*).

Comparing the anterior and posterior regions of the epiblast in sections stained with the above markers did not reveal any regional difference in expression of PAR3 or PKCζ (electronic supplementary material, figure S2A) or between the AP and the AO (electronic supplementary material, figure S2B). However, in whole mounts of embryos immunostained for the Rho family small GTPases RAC1 and RHOA, we observed much stronger expression in cell membranes of the posterior region than in the anterior AP (figure 2*x–ai*). This differential staining increased dramatically with developmental stage after EGK X–XI.

SEM analysis of whole-mounted embryos viewed from the ventral side revealed the gradual formation of the hypoblast sheet during development (figure 2*aj,ak*). At EGK X–XI, cells of posteriorly located hypoblast islands are flattened and extend filopodial extensions to their neighbours (figure 2*aj″*)

while anterior hypoblast cells are spherical and remain as individuals, which do not seem to interconnect by projections (figure 2*aj′*). At EGK XII when the hypoblast sheet has extended about half-way across the embryo, the hypoblast sheet that has formed posteriorly consists of flat and elongated cells with numerous projections and interconnected by numerous filopodia and lamellipodia (figure 2*ak′,ak″*).

Together, the above results show that cell polarization occurs gradually in the epiblast of the AP (but not in the AO) and reveals that the epiblast becomes a columnar epithelium with fully polarized cells before primitive streak formation begins at stage XIV-2. Cell polarization is biased towards the posterior AP not only within the epiblast but also in the lower layer.

## 3.2. Identification of regional markers of the pre-streak embryo by RNAseq

For an initial survey of regionally expressed mRNAs in the chick blastoderm prior to primitive streak formation, we dissected 12 regions, carefully guided by defined anatomical landmarks. We chose the posterior (pAP), anterior (aAP) and whole (AP) area pellucida epiblast (the latter including tissue collected as 'anterior' and 'posterior'), posterior (pMZ), anterior (aMZ) and whole (MZ) marginal zone (the latter including posterior and anterior MZ), posterior (pAO), anterior (aAO) and whole (AO) area opaca (including

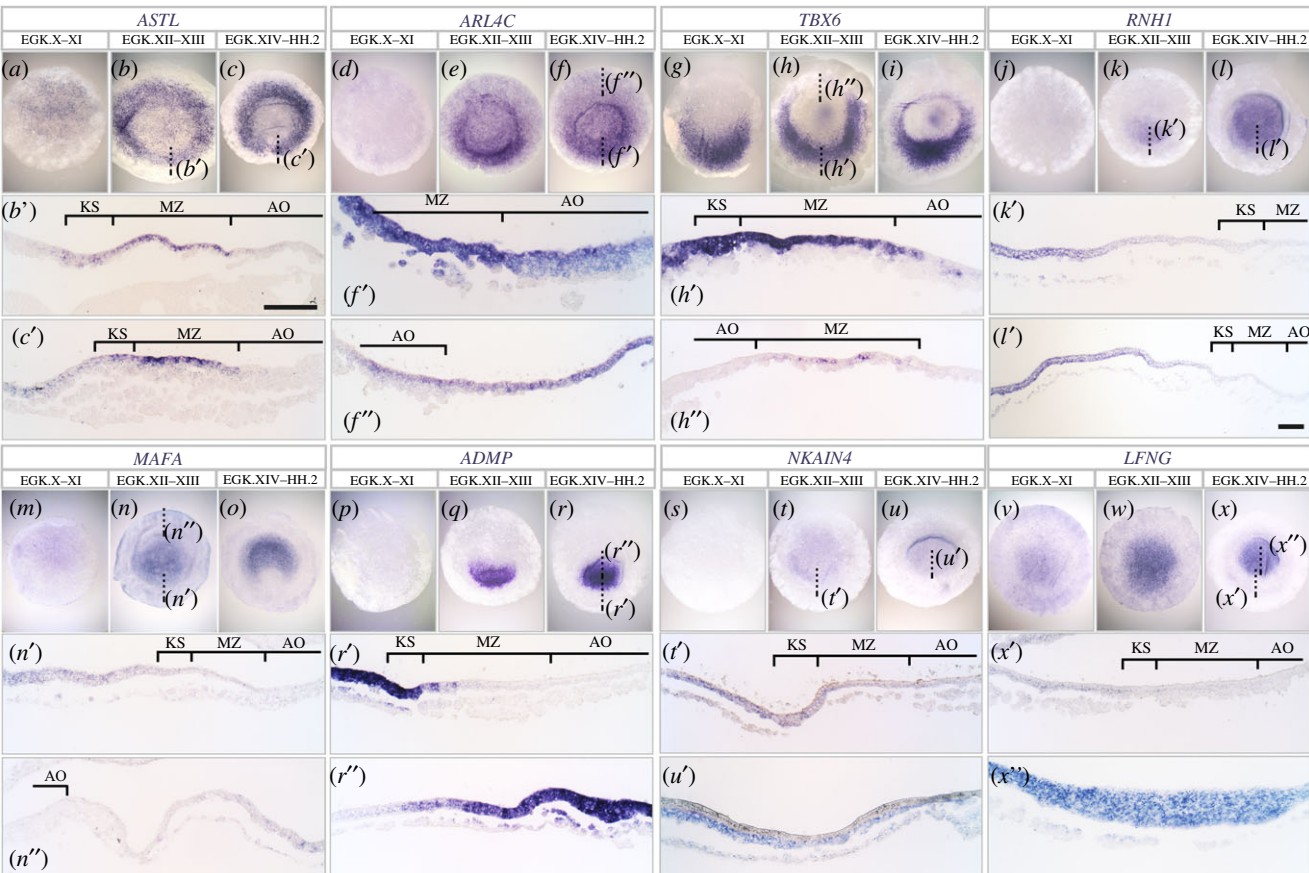

**Figure 3.** Expression of selected genes enriched in MZ and AP. Expression pattern of MZ- (*a–i*) and AP- (*j–x*) enriched genes revealed by *in situ* hybridization in whole-mount and sagittal sections at three stages: EGK X–XI, EGK XII–XIII and EGK XIV–HH.2. Embryonic regions indicated by brackets. The dotted lines show the level of the section. Whole embryos are shown anterior side-up; sections are shown posterior to the right. Scale bars for sections: 100 µm (all sections at the same magnification except (*l′*)).

both epiblast and deep yolky cells of the GW), the germ wall (GW), the hypoblast (Hypo) and KS (electronic supplementary material, figure S3; see also Material and methods section). A hierarchical clustering heat map and principal component analysis (PCA) (electronic supplementary material, figure S3B,C) show that anatomically similar regions cluster together and exhibit more similar expression profiles. The anterior and posterior sub-regions appear distinct in both MZ and AP, while those of the AO show less regional difference. The hypoblast is the most divergent tissue.

To identify putative regional markers, normalized fragments per kilobase of a transcript, per million mapped reads (FPKM) values for each region were compared to those of the other regions. For easy reference, an Excel table summarizing all results is included as electronic supplementary material. From these comparisons, 33 region-enriched genes were selected (electronic supplementary material, tables S3–S8) for further study by *in situ* hybridization and histological sectioning to find the most diagnostic genes for a specific region. The expression patterns of 16 representative genes are illustrated in figures 3 and 4, and the remaining 17 genes are shown in electronic supplementary material, figure S4. Among the three MZ enriched genes, *ASTL* shows the most distinct expression all around the MZ (but extending into the aAP from EGK XII; figure 3*a–c*). *ARL4C* shows similar strong expression in the MZ with lower expression throughout the rest of the blastoderm (figure 3*d–f*). Anteriorly, a stronger expression is seen in the AP than in the MZ (figure 3*f*, *f′*). *TBX6* is also strongly expressed in the

MZ, but less in the anterior side (figure 3*g–i*). All three genes are expressed mainly in the epiblast layer including the epiblast overlying KS.

Five AP-enriched genes are expressed exclusively in the AP (figure 3*j–x*). *RNH1* and *ADMP* show initial strong expression in the posterior AP and then expand anteriorly (figure 3*j–l* and *p–r*). *RNH1* is excluded from the epiblast overlying KS but *ADMP* is expressed there (figure 3*k′,l′,r′*). On the other hand, *MAFA*, *NKAIN4* and *LFNG* are expressed in the entire AP (figure 3*m–o* and *s–x*). Expression of all three genes is mainly in the epiblast or newly formed middle layer cells (figure 3*n′,n″,t′,x′,x″*), but the Na/K-ATPase-associated *NKAIN4* shows strong expression in the middle layer at EGK XIV–HH2 (figure 3*u′*).

For the AO and GW, we noted that all AO enriched genes from our analysis have higher expression in the GW than in the overlying AO epiblast (electronic supplementary material, table S5). Four genes, *DLL1*, *DOC2B*, *DKK1* and *WNT8C* show strong expression in the AO, especially in its lower layer (GW) (figure 4*a–l*). *DLL1* is expressed in the posterior, but not anterior GWM (figure 4*b′,b″*), while *DOC2B* is restricted to the GW (figure 4*e′,e″*). *DKK1* shows strong expression in KS (not including the epiblast above it) as well as in the hypoblast and GW (figure 4*g′-h′*). *WNT8C* is expressed weakly in the MZ and strongly in the GWM but not in the KS region (figure 4*k′* and *k″*).

Among the KS-enriched genes, *PITX2* and *CHRD* have a strong expression in that region (figure 4*m–r*). *PITX2* is expressed as a wide crescent shape in the epiblast at an earlier

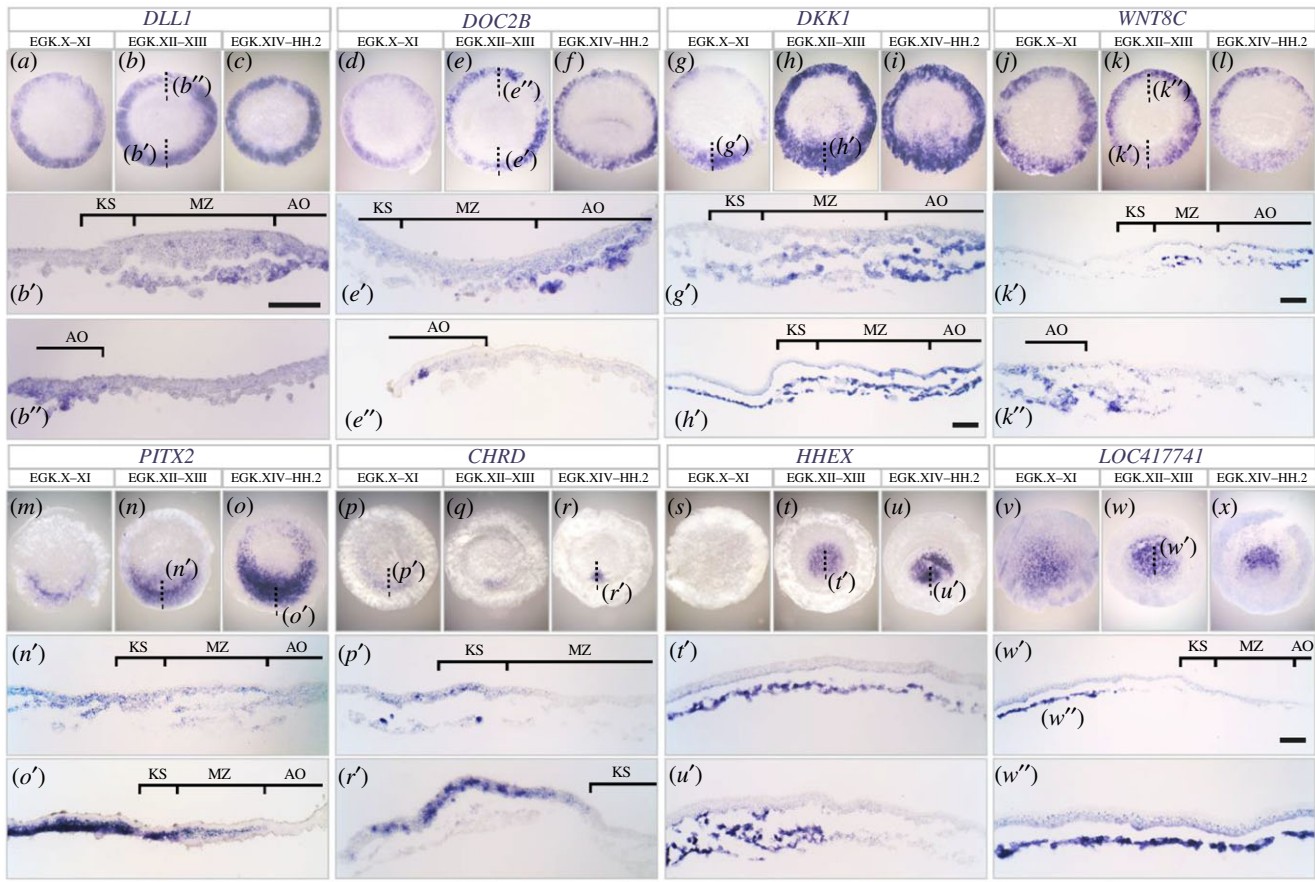

**Figure 4.** Expression of selected genes enriched in AO, GW, KS and hypoblast. Expression pattern of AO- (*a–f*), GW- (*g–l*), KS- (*m–r*) and hypoblast- (*s–x*) enriched genes. Details as in figure 3. Scale bars for sections: 100 μm (all sections at the same magnification except (*h′*), (*k′*) and (*w′*)).

stage (figure 4*n′*) but later in the lower layers (figure 4*o′*). *CHRD*, compared with *PITX2*, has more restricted expression (figure 4*p–r*). Initially, it is expressed in a small region of the pAP and the KS (figure 4*p′*), but soon becomes restricted to the pAP (figure 4*r′*).

Both hypoblast-enriched genes, *HHEX* and *LOC417741* (*secreted frizzled-related protein 2-like*) are expressed only in the hypoblast and therefore become absent posteriorly at EGK XIV–HH2 when the hypoblast is displaced by the encroaching endoblast (figure 4*s–x*). *LOC417741* is expressed in the hypoblast islands at EGK X–XI (figure 4*v*).

## 3.3. Defining the MZ

Because of the functional importance of the MZ especially in relation to its influence in determining the position of the primitive streak in the adjacent AP epiblast [10,21,28,30,31,54–58], we sought other criteria by which it can be identified more easily, especially because most genes expressed in this region are also expressed in at least one of the adjacent regions. Given that the MZ does not contribute cells to the embryo proper (figure 1 and [21]), and since it is a region where cells form a barrier for movements into and out of the AP, we reasoned that there might be characteristic cell shapes that reflect this relatively rigid arrangement and resorted to RAC1 staining to reveal cell peripheries (figure 2). First, the MZ region was marked by the expression of *ASTL* revealed by whole-mount *in situ* hybridization; then, cell shapes were revealed by RAC1 immunostaining (figure 5; electronic supplementary material, figure S5). We found that in the pMZ, cells are elongated circumferentially (at a right angle to the

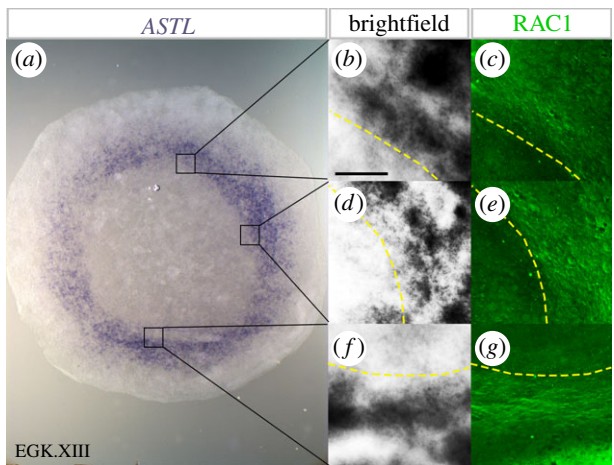

**Figure 5.** Cell polarization in the MZ. Double staining for ASTL mRNA (*a*, *b*, *d*, *f*) and RAC1 immunostaining (*c*, *e*, *g*) reveals a distinctive pattern of cell polarization in the MZ. Elongated cell morphology is observed in all regions of the MZ, including posterior (*f,g*), lateral (*d,e*), and anterior (*b,c*) regions. Yellow dashed lines indicate the boundary between MZ and AP. Scale bar: 100 μm.

radius of the blastodisc) in a distinctive way, not seen in other regions (electronic supplementary material, figure S5). Importantly, this cell orientation in the MZ was observed not only posteriorly (figure 5*a,f,g*), but all around the MZ including lateral (figure 5*d,e*) and anterior (figure 5*b,c*) regions. However, we noted that while the boundary of cell elongation (as seen by RAC1 staining) corresponds quite precisely to anatomical landmarks such as the position of KS, the

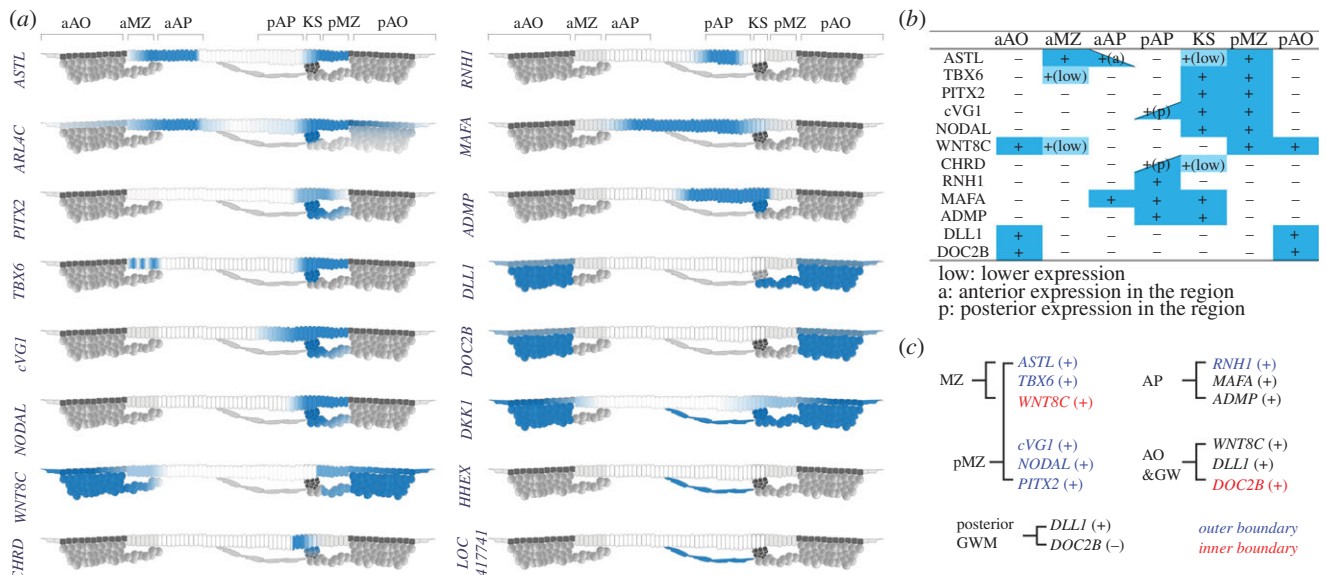

**Figure 6.** Summary of regional markers of the pre-streak chick embryo. (*a*) Schematic summary of gene expression patterns in sections. (*b*) A tabular summary of gene expression patterns in the epiblast of different embryonic regions.+and—indicate presence or absence (or very low) expression, respectively. (*c*) Possible combinations of genes marking specific embryonic regions. Blue and red indicate gene expression domains respecting the outer or inner boundary of the stated region, respectively.

expression of ASTL is not as precisely localized: it displays some heterogeneity especially near its inner and outer boundaries, and some cells expressing this marker are seen outside of the anatomical MZ (figure 5).

# 4. Discussion

## 4.1. Progressive cell polarization from posterior to anterior

The literature on cell morphology in the epiblast has generally described that cells become progressively more columnar as development proceeds between laying (EGK X) and primitive streak formation (HH2) [52,59,60]. Here we show that this process is strongly polarized, like the embryo, from posterior to anterior. One possibility, by analogy to some genes previously described to follow a similar wave-like expression (such as OTX2, GNOT1, GNOT2, PRICKLE1 and CELSR1 (also known as cFMI1)), is that the process is influenced by signals from the underlying hypoblast as this becomes a continuous layer [5,33,61–63]. We also noted that cell elongation is an earlier sign of this process and that it is later followed by segregation of the classical epithelial cell polarity markers including the position of the Golgi apparatus and apical junctions.

## 4.2. Anatomical and functional embryonic regions and their relationships to gene markers

Our RNAseq analysis, clustering and PCA results describe a series of new molecular markers that characterize various embryonic regions at the molecular level (figure 6). For example, the MZ and KS, which have been studied extensively as distinct functional regions [20,21,30,31,33,34,54,57], are identified by characteristic gene expression profiles. However, it is particularly worth noting that the correspondence between gene expression patterns and either the anatomical

features or functional properties of some regions is not perfect. Gene expression domains such as that of ASTL (predominantly expressed in the MZ, all around the embryo) are also seen in patches of cells both inside the AP and more externally in the AO. No transcripts were found to be entirely specific to the MZ. However, the anatomical MZ does appear to correspond quite closely to a region of cell elongation (at right angles to the radius of the blastoderm) revealed by RAC1 staining. These observations raise the interesting question of whether gene expression or anatomical and functional features (including cell fate and inducing abilities) should be dominant in defining embryonic regions and has important implications for the interpretation of experiments that attempt to identify cell identities (or even cell fates) by single-cell RNAseq.

Until now, no markers for the anterior MZ (aMZ) had been reported. This is not because the anterior part of the embryo does not have distinct molecular identity since BMP-signalling and some of the targets of this pathway such as GATA2 are strongly enriched in anterior components (aAO, aMZ and aAP) [64,65]. However, even our systematic search for genes expressed specifically in the MZ did not identify components that are specific and exclusive to the MZ. Interestingly, those that mark the posterior and postero-lateral MZ such as ASTL and ARL4C are not expressed the same way as the MZ is defined in the literature (a lateral-anterior projection of the area defined as pMZ). Rather, the more anterior the position, the more central (AP-biased) the expression of these markers becomes especially ASTL. ASTL encodes an astacin-like protease that may be involved in the processing/maturation of BMP-related factors, so it is possible that this is an important feature of its expression, but it also raises the possibility that the MZ as such is not definable either anatomically or molecularly for anterior regions of the embryo and that genes like ASTL, expressed in the pMZ posteriorly, cannot be extrapolated all around the circumference. However, functional experiments do suggest that a distinct aMZ must exist. For example, misexpression of cVG1 in outside the MZ has no

royalsocietypublishing.org/journal/rsob    Open Biol. **10**: 190299

effect, whereas when misexpressed in the aMZ it can induce a complete axis. This difference has been ascribed to Wnt activity [57]. Therefore, as for the pMZ, the spatial localization of functional and (at least some) molecular properties do not appear to be identical.

Overall, this study provides a good resource to define and study regions based on their anatomical, functional and molecular properties. However, in addition to the problem of these markers extending to some neighbouring regions in some cases (discussed above), it is also clear that a molecular definition of most regions requires consideration of the expression of more than one marker.

At these early stages, the discrepancy between anatomy, functional properties and molecular signatures seems more extensive than at later stages, when tissues have started to diverge functionally and molecularly by a much greater extent. But it raises the general question of which of these three criteria (anatomical definition, spatio-temporal map of particular functions and gene expression territories) should be the dominant feature used to refer to particular regions. We suggest that all three be taken into account, but that the classical anatomical names remain used as they always have been so that important findings from the older literature can be understood by future generations. These considerations bring to the fore the motto that 'genes mark cell states rather than cell fates' [34,66], which was proposed after observing that cells moving around the embryo entered and left domains of gene expression and changed their expression according to their current location. While there is indeed a 'molecular anatomy', this does not always neatly overlap with the real anatomy (morphological or functional) of the embryo.

Despite these thoughts, it is also clear that some regions are indeed characterized by the overlap of a number of genes with very important functions, which define the properties of that region. A clear example at this early stage is the pMZ, where several secreted factors are co-expressed, including cVG1, NODAL (electronic supplementary material, figure S6) and WNT8C, along with a set of important transcription factors (TBX6, PITX2) (figure 6). Our current (figure 6) and previous [58] findings reveal further co-expressed genes that overlap in posterior regions which may also be involved in a complex set of events that position the primitive streak as the site of gastrulation. However, although the pMZ does have unique properties of being able to induce this without contributing cells to the embryo proper (the properties that defined the Nieuwkoop Centre of amphibian embryos) [21], it is difficult at this point to find clear molecular correlates of this that do not overlap into adjacent regions. It is clear, however, that it is combinations of genes, rather than single key markers, that characterize the complex topography and the spatio-temporal distribution of functional properties in embryos at this stage of development.

# 5. Material and methods

## 5.1. Tissue sampling, mRNA extraction and RNAseq

Fertilized White Leghorn hens' eggs (Henry Stewart, UK) were incubated for 6–8 h to obtain EGK XII–XIII embryos. A total of 12 different regions was excised in PBS, collected in RNALater (Invitrogen), and stored at 4°C until mRNA

purification. The regions collected are shown diagrammatically in electronic supplementary material, figure S3. For all AO, MZ, AP, and including their anterior or posterior parts, only the epiblast layer was collected. In the case of the AO, the deep GW was removed as much as possible, but it was not possible to do this completely because of its strong attachment to the overlying AO epiblast. The AO did not include the extreme marginal cells (margin of overgrowth) [67–70]. For AP, the hypoblast (islands and sheet) was removed. For KS, all cell layers, including epiblast, middle layer (KS itself) and endodermal layer, were collected. After collection, mRNA was purified using the RNeasy Mini Kit (Qiagen) according to the manufacturer's instructions. Purified mRNA was titrated using Aglient 2200 Tapestation. The number of embryos, concentration of mRNA and RNA integrity number are listed in electronic supplementary material, table S1. Libraries were constructed from 100 ng total RNA using the NEBNext Ultra™ II Directional RNA Library Prep Kit with NEBNext Poly(A) mRNA Magnetic Isolation Module (NEB E7760 and E7490) according to the manufacturer's instructions. Libraries to be multiplexed in the same run were pooled in equimolar quantities, calculated from Qubit and Bioanalyzer fragment analysis. Samples were sequenced with the Illumina NextSeq 500 System (San Diego, US) which gave a total of more than 550 million reads (on average, 47 million reads per sample). The raw data files have been deposited in ArrayExpress 'Anatomical and molecular dissection of the pre-streak chick embryo', accession number E-MTAB-7941.

## 5.2. RNAseq and analysis

Sequence reads were trimmed using trimmomatic 0.36 [71] and aligned to the galGal5 version of the chicken genome using TopHat2 [72]; alignment rates were 83.9% ± 0.8%. Transcripts were counted and normalized using Cufflinks programmes Cuffquant and Cuffnorm, respectively [73]. Data analysis was performed in the R environment. The matrix of transcript FPKMs, which contains an expression of 15 543 genes in 12 samples, was used to calculate differential gene expression. To select candidate markers for a region in the upper cell layer of the pre-primitive-streak stage embryo, FPKM ≥ 10 and fold change (FC) were used for filtering differentially expressed genes, choosing the most appropriate or comparable tissues. For example, the MZ was compared to its neighbouring regions AO and AP, and the aMZ and pMZ to each other and to their corresponding neighbouring regions. The FC cut-off value for each region was adjusted based on the length of the candidate gene list. The cut-offs are given in the supplementary tables together with candidate genes. To identify markers for regions in the lower cell layer, including Hypoblast, GW and KS, candidate genes were selected from the comparisons between a sample to the other 11 samples with cut-off FPKM ≥ 10 and FC given in the supplementary tables (except for the KS candidates that do not require the comparison with GW). The FPKMs of the top 2000 most variable genes, which have the largest squared standard deviation values across 12 samples, were used for the hierarchical clustering heat map and PCA using R (electronic supplementary material, figure S3B,C). The full dataset has been deposited in ArrayExpress 'Anatomical and molecular dissection of the pre-streak chick embryo', accession number E-MTAB-7941. In addition, an

Excel table summarizing all results is included as electronic supplementary material with this paper, for easy reference.

## 5.3. Lineage tracing

Fertilized White Leghorn hens' eggs (Henry Stewart, UK) were incubated for 2–4 h to harvest EGK XI embryos, which then were set-up for modified New culture [23,74]. All manipulations were conducted in Pannett–Compton saline [75]. To label cells, embryos were turned over in a Petri dish to expose the target cell layer and labelled with the carbocyanine dye DiI (1,1′dioctadecyl-3,3,3′,3′-tetra-methyl indocarbocyanine perchlorate, Molecular Probes Inc.) with a modification of previously described methods [19,34,76]. The dye solution was prepared by mixing 0.5 µl 2 mM DiI, 8.5 µl 0.3 M sucrose and 1 µl 0.01% Tween 20 (Sigma). A small volume of dye solution in a microneedle was applied to target positions by using a mouth pipette with gentle air pressure.

## 5.4. SEM and image analysis

For SEM, embryos were cut either sagittally or transversely, then fixed in 3% glutaraldehyde in 0.1 M Na-cacodylate buffer at 4°C overnight. They were washed three times for 10 min each with 0.1 M sodium cacodylate buffer, soaked in 1% osmium tetroxide at 4°C for 30 min, and dehydrated serially in 25, 50, 75 and 100% EtOH for 5 min each. Then, samples were dried with a critical point dryer and mounted on an aluminium block. Finally, they were Au/Pd-coated before imaging. The samples were imaged in a Jeol JSM 7401F field emission scanning electron microscope operated at 2 kV acceleration voltage, beam current at 10 µA, under high vacuum collecting the secondary electrons. Image analysis was conducted using Fiji software [77]. Serial images of the embryo were combined using the pairwise stitching plugin [78]. Each cell was defined using the region of interest (ROI) manger by drawing the outline of the cell-edge manually. The AR of each cell was measured and this converted to a heat map using the ROI colour coder plugin [79], with the min-max range set as 1–6. The measured ARs were exported to Excel for further analysis.

## 5.5. Whole-mount *in situ* hybridization and immunohistochemistry

Whole-mount *in situ* hybridization was performed as previously described [80,81]. The sources of probes used are listed in electronic supplementary material, table S9; some were generated from the chick EST collection [82]. Stained embryos were imaged on an Olympus SZH10 stereo-microscope with a QImaging Retiga 2000R camera. Immuno-histochemistry was performed as previously described [83]. The following antibodies were used: GM130 (BD Biosciences, Cat. 610822), PAR3 (Millipore 07-330), PKCζ (Santa Cruz SC-216), RAC1 (Millipore 05-389), RHOA (Santa Cruz SC-179), Alexa Fluor 488-conjugated goat anti-mouse IgG (Invitrogen A21202), Alexa Fluor 488-conjugated goat anti-rabbit IgG (Invitrogen A11008), Cy3-conjugated anti-mouse IgG (Jackson 115-165-073), all diluted 1:200. For phalloidin staining after immunohistochemistry, embryos were incubated with rhodamine-phalloidin (Invitrogen R415) diluted 1:40 in PBST overnight at room temperature. After incubation, embryos were washed three times in PBS and mounted with Prolong Gold anti-fade reagent with 4′,6-diamidino-2-phenylindole (Invitrogen P36931) as a nuclear counterstain. The stained embryos were imaged with a Leica SPE1 confocal microscope. For sectioning, fixed embryos were embedded in paraffin and sectioned at 10 µm).

## 5.6. Statistical analysis

Statistical analyses were conducted using the GraphPad Prism 7 program. AR values of the same regions but different embryos were pooled and statistically analysed by one-way ANOVA followed by Tukey's post-test for comparison among multiple groups and by the unpaired Student's *t*-test for comparison between two groups.

Data accessibility. All data underlying this paper have been deposited in publicly accessible sites as follows: the full RNAseq dataset has been deposited in ArrayExpress 'Anatomical and molecular dissection of the pre-streak chick embryo', accession number E-MTAB-7941. For easy reference, an Excel file containing comparisons of gene expression in all regions is included as electronic supplementary material. *In situ* hybridization results that are not included in the main figures of the paper are included as electronic supplementary material on the journal's website. A table of the sources of cDNA probes including sequence information (either as accession numbers or references to published papers) is also included in the electronic supplementary material associated with this paper. cDNAs used to make the probes used in this paper are available from the authors on request.

Authors' contributions. H.C.L.: performed experiments and wrote the paper. H.-C.L.: performed bioinformatic analysis. M.T.: performed SEM. N.M.M.O. and Y.Y.: assisted with molecular biology. I.D.A.: performed some *in situ* hybridization. C.D.S.: directed the study, obtained funding and wrote the paper.

Competing interests. We declare we have no competing interests.

Funding. This research was supported by Basic Science Research Program through the National Research Foundation of Korea (NRF) funded by the Ministry of Education (grant no. 2014R1A6A3A 03053468) to H.C.L., and by a Wellcome Trust Investigator Award (grant no. 107055/Z/15/Z) to C.D.S.

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
