## [Reviewer comments · Open Biology]

Review History

RSOB-19-0299.R0 (Original submission)

Review form: Reviewer 1

Recommendation

Major revision is needed (please make suggestions in comments)

Do you have any ethical concerns with this paper?

No

Comments to the Author

In this study, the authors provide a concise summary of the anatomy and terminology relevant to pre-primitive streak avian embryology, and provide a unified nomenclature for future studies. The main analyses in this study involve SEM and immunohistochemistry to characterize cellular organization and polarization, followed by differential RNAseq and in situ hybridization to describe 33 gene expression patterns that define the relevant domains. This study provides a careful characterization of the early embryo, but does not include any functional validation or hypothesis testing that would elevate it to be of broader interest.

While the SEM analysis presented in Figure 2 is compelling, the complementary studies in cell polarity markers lacks the same amount of rigor. The authors fail to quantitate their immunohistochemistry experiments, and present their results in someone obscure panels. In

particular, the conclusions based on Par3 immunostaining is difficult to corroborate for the reader without quantitation or a single-channel display (the DAPI obscures interpretation of the dim Par3 signal).

The authors note in their discussion that cell elongation in the AP precedes the segregation of epithelial cell polarity markers; however, they do not elaborate on this point to describe the implications of this important observation.

The authors then proceed to perform transcriptional profiling in the various tissues at this early stage, and validate the expression profiles for 33 of the differentially-expressed genes. This is a valuable resource, which is summarized well in Figure 6. To increase ease of interpretation, I would recommend adding a summary of the expression domains for each gene as predicted by RNAseq to Figure 3 and Figure 4.

Finally, the authors provide a complete description of the expression patterns of their tested genes; however, they do not discuss the possible implications of these expression patterns for the majority of the displayed genes.

In general, the study is useful but rather difficult to read. Removing too frequently used abbreviations might help considerably.

Review form: Reviewer 2

Recommendation

Accept with minor revision (please list in comments)

Do you have any ethical concerns with this paper?

No

Comments to the Author

This paper presents a collection of resources for studying the chick pre-gastrulation stage embryo. The chick (and more broadly avian) embryo has been a basic system for studying vertebrate embryology for 200 years and more, and many basic embryological discoveries have been made over the years in the avian system. Due to its flat morphology, it also serves in several ways as a better model for the human embryo than does the mouse model, and thus is very useful for understanding human development. Over the years, a range of terminologies has been used for describing features of the early chick embryo. This multiplication of terminologies has added confusion to the field, making it difficult to compare between studies. One of the very valuable features of the current paper is that it presents a correlative and unifying Table, diagram, and discussion of these various terminologies. I anticipate that this summary, prepared by the laboratory of Claudio Stern, who truly knows the early chick embryo and its extensive experimental history, will become the “gold standard” to which future studies will refer, and which will extend the usefulness of classical studies.

The paper presents an RNAseq study of 12 regions of the pre-streak embryo, selected for their interest in the context of previous studies of early avian development. This database will be deposited in public realm, and should be a very useful resource for the developmental biology community. Many important structures have been studied in chick as well as in amphibian, fish, and mouse embryos. The current data will facilitate molecular comparisons between “equivalent” regions in different embryos (e.g. chick GWM vs. amphibian Nieuwkoop center; chick hypoblast vs. mouse anterior visceral endoderm), thus allowing for coming to general conclusions. In studies included in the paper, the authors find that RNA's that are relatively specific to a specific region nevertheless do not align precisely with the borders of the that region as anatomically defined. In an interesting and important discussion, the authors warn against the

blind assignment of gene expression patterns to anatomical features, as is sometimes done in RNAseq experiments, emphasizing that an understanding of the development of a tissue or region requires integrating information on multiple levels.

Also presented in the paper is a set of useful novel data regarding epithelialization of the blastodisc with regards to its anterior-posterior axis.

Overall, this is a concise, well-informed paper containing several elements that should serve as a valuable resource for the avian and broader embryological communities. The figures are very nice and clear, and easy to understand.

Specific comments:

1. The authors selected 33 genes from the RNAseq analysis for analysis by in situ hybridization, and they present 16 of these in the paper. It is not clear what criteria were used to select these 16. I would encourage the inclusion of the data regarding the other 17 genes in the supplementary material. Assuming the expression patterns are informative, this would increase the usefulness of the resource.

1. In Figure 2 L,O,R, the paper claims that Par3 undergoes progressive apical localization. I was not convinced by the data. Par3 appears to be localized at both the apical and basal surfaces at all stages. This is in contrast to PKC, which does appear to undergo relocalization.

2. In the Methods section, it should be stated what were the conditions of SEM image acquisition.

Decision letter (RSOB-19-0299.R0)

08-Jan-2020

Dear Professor Stern,

We are pleased to inform you that your manuscript RSOB-19-0299 entitled "Molecular anatomy of the pre-primitive-streak chick embryo" has been accepted by the Editor for publication in Open Biology. The reviewer(s) have recommended publication, but also suggest some minor revisions to your manuscript. Therefore, we invite you to respond to the reviewer(s)' comments and revise your manuscript.

Please submit the revised version of your manuscript within 7 days. If you do not think you will be able to meet this date please let us know immediately and we can extend this deadline for you.

Please see our detailed instructions for revision requirements <https://royalsocietypublishing.org/journals/authors/author-guidelines/>.

- 1) A text file of the manuscript (doc, txt, rtf or tex), including the references, tables (including captions) and figure captions. Please remove any tracked changes from the text before submission. PDF files are not an accepted format for the "Main Document".
- 2) A separate electronic file of each figure (tiff, EPS or print-quality PDF preferred). The format should be produced directly from original creation package, or original software format. Please note that PowerPoint files are not accepted.
- 3) Electronic supplementary material: this should be contained in a separate file from the main text and meet our ESM criteria (see <https://royalsociety.org/journals/authors/author-guidelines/>). All supplementary materials accompanying an accepted article will be treated as in their final form. They will be published alongside the paper on the journal website and posted on the online figshare repository. Files on figshare will be made available approximately one week before the accompanying article so that the supplementary material can be attributed a unique DOI.

Online supplementary material will also carry the title and description provided during submission, so please ensure these are accurate and informative. Note that the Royal Society will not edit or typeset supplementary material and it will be hosted as provided. Please ensure that the supplementary material includes the paper details (authors, title, journal name, article DOI). Your article DOI will be 10.1098/rsob.2016[*last 4 digits of e.g. 10.1098/rsob.20160049*].

- 4) A media summary: a short non-technical summary (up to 100 words) of the key findings/importance of your manuscript. Please try to write in simple English, avoid jargon, explain the importance of the topic, outline the main implications and describe why this topic is newsworthy.

Images

Data-Sharing

It is a condition of publication that data supporting your paper are made available. Data should be made available either in the electronic supplementary material or through an appropriate repository. Details of how to access data should be included in your paper. Please see <https://royalsociety.org/journals/authors/author-guidelines/> for more details.

Data accessibility section

Sincerely,

The Open Biology Team

<mailto:openbiology@royalsociety.org>

Reviewer(s)' Comments to Author:

Referee: 1

Comments to the Author(s)

In this study, the authors provide a concise summary of the anatomy and terminology relevant to pre-primitive streak avian embryology, and provide a unified nomenclature for future studies. The main analyses in this study involve SEM and immunohistochemistry to characterize cellular organization and polarization, followed by differential RNAseq and in situ hybridization to describe 33 gene expression patterns that define the relevant domains. This study provides a careful characterization of the early embryo, but does not include any functional validation or hypothesis testing that would elevate it to be of broader interest.

While the SEM analysis presented in Figure 2 is compelling, the complementary studies in cell polarity markers lacks the same amount of rigor. The authors fail to quantitate their immunohistochemistry experiments, and present their results in someone obscure panels. In particular, the conclusions based on Par3 immunostaining is difficult to corroborate for the reader without quantitation or a single-channel display (the DAPI obscures interpretation of the dim Par3 signal).

The authors note in their discussion that cell elongation in the AP precedes the segregation of epithelial cell polarity markers; however, they do not elaborate on this point to describe the implications of this important observation.

The authors then proceed to perform transcriptional profiling in the various tissues at this early stage, and validate the expression profiles for 33 of the differentially-expressed genes. This is a valuable resource, which is summarized well in Figure 6. To increase ease of interpretation, I would recommend adding a summary of the expression domains for each gene as predicted by RNAseq to Figure 3 and Figure 4.

Finally, the authors provide a complete description of the expression patterns of their tested genes; however, they do not discuss the possible implications of these expression patterns for the majority of the displayed genes.

In general, the study is useful but rather difficult to read. Removing too frequently used abbreviations might help considerably.

Referee: 2

Comments to the Author(s)

This paper presents a collection of resources for studying the chick pre-gastrulation stage embryo. The chick (and more broadly avian) embryo has been a basic system for studying vertebrate embryology for 200 years and more, and many basic embryological discoveries have been made over the years in the avian system. Due to its flat morphology, it also serves in several ways as a better model for the human embryo than does the mouse model, and thus is very useful for understanding human development. Over the years, a range of terminologies has been used for describing features of the early chick embryo. This multiplication of terminologies has added confusion to the field, making it difficult to compare between studies. One of the very valuable features of the current paper is that it presents a correlative and unifying Table, diagram, and discussion of these various terminologies. I anticipate that this summary, prepared by the laboratory of Claudio Stern, who truly knows the early chick embryo and its extensive experimental history, will become the "gold standard" to which future studies will refer, and which will extend the usefulness of classical studies.

The paper presents an RNAseq study of 12 regions of the pre-streak embryo, selected for their interest in the context of previous studies of early avian development. This database will be deposited in public realm, and should be a very useful resource for the developmental biology community. Many important structures have been studied in chick as well as in amphibian, fish, and mouse embryos. The current data will facilitate molecular comparisons between "equivalent" regions in different embryos (e.g. chick GWM vs. amphibian Nieuwkoop center; chick hypoblast vs. mouse anterior visceral endoderm), thus allowing for coming to general conclusions. In studies included in the paper, the authors find that RNA's that are relatively specific to a specific region nevertheless do not align precisely with the borders of the that region as anatomically defined. In an interesting and important discussion, the authors warn against the blind assignment of gene expression patterns to anatomical features, as is sometimes done in RNAseq experiments, emphasizing that an understanding of the development of a tissue or region requires integrating information on multiple levels.

Also presented in the paper is a set of useful novel data regarding epithelialization of the blastodisc with regards to its anterior-posterior axis.

Overall, this is a concise, well-informed paper containing several elements that should serve as a valuable resource for the avian and broader embryological communities. The figures are very nice and clear, and easy to understand.

Specific comments:

1. The authors selected 33 genes from the RNAseq analysis for analysis by in situ hybridization, and they present 16 of these in the paper. It is not clear what criteria were used to select these 16. I would encourage the inclusion of the data regarding the other 17 genes in the supplementary material. Assuming the expression patterns are informative, this would increase the usefulness of the resource.

1. In Figure 2 L,O,R, the paper claims that Par3 undergoes progressive apical localization. I was not convinced by the data. Par3 appears to be localized at both the apical and basal surfaces at all stages. This is in contrast to PKC, which does appear to undergo relocalization.

2. In the Methods section, it should be stated what were the conditions of SEM image acquisition.

Author's Response to Decision Letter for (RSOB-19-0299.R0)

See Appendix A.

Decision letter (RSOB-19-0299.R1)

13-Jan-2020

Dear Professor Stern,

We are pleased to inform you that your manuscript entitled "Molecular anatomy of the pre-primitive-streak chick embryo" has been accepted by the Editor for publication in Open Biology.

Article processing charge

Please note that the article processing charge is immediately payable. A separate email will be sent out shortly to confirm the charge due. The preferred payment method is by credit card; however, other payment options are available.

Sincerely,

The Open Biology Team
mailto: openbiology@royalsociety.org

Appendix A

Dear Editor,

Thank you very much for your positive decision on this manuscript, with which we are very pleased. We are pleased that the editors as well as both reviewers appreciated the value of our paper. We have now revised the submission in accordance with these recommendations. Specifically, we have:

(1) included a better description of the methods used for SEM imaging in the Materials and Methods;

(2) Included a full "Data Accessibility" section listing the locations of the underlying data and information as well as for availability of the cDNA probes used;

(3) we have added the information requested by the editor (DOI etc.) to the front of the Supplementary Information

(4) we have now included an Excel file (as Electronic Supplementary Material) containing all RNAseq results in all samples, which allows searching and direct comparisons, for ease of reference.

Here are responses to other points raised by the reviewers:

REFEREE 1:

While the SEM analysis presented in Figure 2 is compelling, the complementary studies in cell polarity markers lacks the same amount of rigor. The authors fail to quantitate their immunohistochemistry experiments, and present their results in someone obscure panels. In particular, the conclusions based on Par3 immunostaining is difficult to corroborate for the reader without quantitation or a single-channel display (the DAPI obscures interpretation of the dim Par3 signal).

RESPONSE: We feel that a full quantitative analysis of the progress of epithelialisation based on many markers is beyond the scope of this paper. We decided to include a description of the overall changes because we felt that this would provide useful reference points for readers to interpret the expression of various markers, but the paper is not focused on the process of epithelialisation itself.

The authors note in their discussion that cell elongation in the AP precedes the segregation of epithelial cell polarity markers; however, they do not elaborate on this point to describe the implications of this important observation.

RESPONSE: as this is a report in the journal's "Methods and Techniques" section, to provide a reference resource for the community, we opted to minimise all speculation and to restrict ourselves to completely factual descriptions as much as possible. For this reason our Discussion section is short and restricted to pointing out the most important/salient findings from the Results.

The authors then proceed to perform transcriptional profiling in the various tissues at this early stage, and validate the expression profiles for 33 of the differentially-expressed genes. This is a valuable resource, which is summarized well in Figure 6. To increase ease of interpretation, I would recommend adding a summary of the expression domains for each gene as predicted by RNAseq to Figure 3 and Figure 4.

RESPONSE: unfortunately this is not really possible - the RNAseq results are just for 12 specific regions, and therefore not a "spatial map" as such (as per in situ hybridisation). To improve the usefulness of the paper, we have now included an easy to search Excel table as Electronic Supplementary Material. Any reader can easily compare the expression levels of any gene across all 12 regions from this table, as well as compare different genes to each other.

Finally, the authors provide a complete description of the expression patterns of their tested genes; however, they do not discuss the possible implications of these expression patterns for the majority of the displayed genes.

RESPONSE: as pointed out above, this is a descriptive paper specifically to identify regional markers and to define anatomical regions of these embryos. As appropriate for a paper in the "Methods and Techniques" section of the journal, we felt we should refrain from too much speculation about function of the genes or other issues. The results presented will open up many questions and it will be up to the community to pick up on the many interesting issues to address in future studies.

In general, the study is useful but rather difficult to read. Removing too frequently used abbreviations might help considerably.

RESPONSE: after going through the manuscript we have opted to keep most of the abbreviations for the sake of brevity - it becomes very awkward to spell out all the long names repeatedly throughout the manuscript.

REFeree 2:

1. The authors selected 33 genes from the RNAseq analysis for analysis by in situ hybridization, and they present 16 of these in the paper. It is not clear what criteria were used to select these 16. I would encourage the inclusion of the data regarding the other 17 genes in the supplementary material. Assuming the expression patterns are informative, this would increase the usefulness of the resource.

RESPONSE: (see above). The referee seems to have missed the supplementary figure that includes these expression patterns. All 33 genes are included in the paper. The most salient, useful or interesting genes were selected for the main part of the manuscript and are summarised in the diagram, but all 33 are included.

2. In Figure 2 L,O,R, the paper claims that Par3 undergoes progressive apical localization. I was not convinced by the data. Par3 appears to be localized at both the apical and basal surfaces at all stages. This is in contrast to PKC, which does appear to undergo relocalization.

RESPONSE: it seems that the reviewer misunderstood our statement, which states that PAR3 does NOT show as strong an apical localisation as the other components listed.

3. In the Methods section, it should be stated what were the conditions of SEM image acquisition.

RESPONSE: this has been done.

=====

We hope that the revisions made will satisfy the editors and look forward to seeing the paper published in Open Biology.

Sincerely

Claudio Stern on behalf of all authors